# An Update on Effectiveness and Practicability of Plant Essential Oils in the Food Industry

**DOI:** 10.3390/plants11192488

**Published:** 2022-09-22

**Authors:** Liana Claudia Salanță, Janna Cropotova

**Affiliations:** 1Department of Food Science, Faculty of Food Science and Technology, University of Agricultural Sciences and Veterinary Medicine Cluj-Napoca, 400372 Cluj-Napoca, Romania; 2Department of Biological Sciences Ålesund, Norwegian University of Science and Technology, Larsgårdsvegen 4, 6025 Ålesund, Norway

**Keywords:** plants, essential oils, bioactive compounds, shelf life, food safety

## Abstract

Consumer awareness and demands for quality eco-friendly food products have made scientists determined to concentrate their attention on sustainable advancements in the utilization of bioactive compounds for increasing safety and food quality. Essential oils (EOs) are extracted from plants and exhibit antimicrobial (antibacterial and antifungal) activity; thus, they are used in food products to prolong the shelf-life of foods by limiting the growth or survival of microorganisms. In vitro studies have shown that EOs are effective against foodborne bacteria, such as *Escherichia coli*, *Listeria monocytogenes*, *Salmonella* spp., and *Staphylococcus aureus*. The growing interest in essential oils and their constituents as alternatives to synthetic preservatives has been extensively exploited in recent years, along with techniques to facilitate the implementation of their application in the food industry. This paper’s aim is to evaluate the current knowledge on the applicability of EOs in food preservation, and how this method generally affects technological properties and consumers’ perceptions. Moreover, essential aspects concerning the limitation of the available alternatives are highlighted, followed by a presentation of the most promising trends to streamline the EOs’ usability. Incorporating EOs in packaging materials is the next step for green and sustainable foodstuff production and a biodegradable method for food preservation.

## 1. Introduction

The technological advancements, globalization, and economic growth of the last decade have led the way to urban expansion, triggering significant changes in consumers’ lifestyles and dietary habits. Nowadays, one of the most important changes in consumers’ lifestyles is increased health consciousness by understanding the importance of food in maintaining and improving human well-being [1,2]. Worldwide, the green earth concept determines the use of natural products in daily life.

Food perishable products are prone to oxidation and microbial contamination, leading to changes in sensorial properties, loss of essential nutrients, and major economic losses for the industry. The addition of artificial preservatives and antioxidants during industrial food production is necessary to reduce all these effects and to improve the safety, shelf-life, and quality of the products. Although synthetic antioxidants, such as butylated hydroxytoluene, butylated hydroxyanisole, and propyl gallate, are cheaper and more stable, concerns were risen about their safety: long-term intake may lead to allergies, gastrointestinal disorders, or even cancer [3]. Similarly, nitrites and benzoates used at high concentrations could be toxic and carcinogenic [4]. There is a clear trend toward finding solutions for natural food protection as aspects such as clean labeling, and healthy or organic food concepts are increasingly demanded by consumers. The main challenges are maintaining consumers’ acceptance, good functionality, and low costs. An alternative solution is the use of valuable compounds from plants. The most common bioactive compounds used for food preservation with proven efficacy against microbial contamination are phenolic compounds and essential oils [5].

Essential oils (EOs), mainly comprising the aromatic and volatile compounds naturally present in all parts of plants, are known as highly effective antimicrobial agents. The antimicrobial (preventing spoilage by inhibiting microorganisms’ growth) and antioxidant (preventing lipid oxidation and peroxidation) activities of EOs are attributed to their volatile compounds; thus, they have been extensively utilized for medicinal, bactericidal, virucidal, fungicidal, antiparasitic, insecticidal, and antioxidant purposes [6,7,8,9,10,11,12]. Primarily used in aromatherapy, EOs have gained the attention of the food industry in the last decade due to their capacity to control the growth of pathogenic microorganisms [6]. The main essential oils with proven antimicrobial properties are obtained from certified crops, such as lavender, thyme, cinnamon, oregano, basil, or rosemary. Essential oils and extracts are also attained from by-products resulting from agricultural wastes such as grape seeds, apricot kernels, and orange peels [13,14,15]. EOs contain from ten to hundreds of individual compounds with 2–3 major bioactive components present at high concentrations (20–70%), including phenylpropanoids (e.g., phenols, aldehydes, and alcohols), terpenoids (phenols, ethers, ketones, aldehydes, alcohols, and esters), and terpene compounds (e.g., limonene, p-cymene, and terpinene). Terpenes or terpenoids are known as active agents against bacteria, fungi, viruses, and protozoa [16]. It is essential to mention that their composition may vary, due to intrinsic factors: depending on the used plant part (roots, stems, leaves, seeds, flowers, and others), and extrinsic factors: extraction method (water or steam distillation, solvent extraction, expression under pressure, microwave-assisted extraction, ultrasonic-assisted extraction, supercritical fluid extraction, and subcritical water extraction) [17]. The steam distillation technique and principal methods of EOs application in the food industry are illustrated in Figure 1.

The mechanism of antibacterial action of EOs has been reported in several studies [6,18,19,20,21]. These oils are complex in nature, with a mixture of over 300 distinct compounds at different concentrations. The major components such as terpenes/terpenoids and aromatic/aliphatic are responsible for the various biological activities, including food preservation [6,22]. Most of the EOs and their constituents have been admitted as safe food additives by the Food and Drug Administration (FDA) [23].

The antibacterial properties of EOs might be attributed to more than one mechanism [24]. Due to the lipophilicity of terpenes and phenolics, EOs are capable of penetrating easily inside the cytoplasm of microorganisms and deteriorating the phospholipid bilayer of mitochondria and the inner membrane, resulting in increased cellular permeability followed by leakage of cytoplasmic constituents (e.g., DNA and RNA) and certain ions (Na^+^, K^+^, and Mg^2+^) [25]. Another mechanism is related to the distortion of lipid–protein interaction in a bacterial cell caused by lipophilic hydrocarbons naturally found in EOs affecting the ATPases activity necessary for the production of ATP [26]. In addition, certain phenolics found in EOs may disrupt the electron flow, proton motive force, and cytoplasmic coagulation. All these mechanisms result in the inhibition of the bacterial activity on the food surface, but also prevent active compounds from EOs from reaching the inner membrane. Therefore, Gram-negative bacteria have been shown to be more resistant to EOs than Gram-positive bacteria [27]. Nevertheless, due to hydrophobic compounds present in EOs that are capable of passing through the barrier of the outer membrane, it is still possible to inactivate Gram-negative bacteria in food products. Thus, Gram-negative bacteria have a lower susceptibility to EOs than Gram-positive ones, mainly due to their membrane characteristics that act as barriers against macromolecules and hydrophobic compounds [28]. The main bioactive compounds with significant antimicrobial activities are terpineol, thujanol, myrcenol, neral, thujone, camphor, and carvone [6].

In addition, essential oils have also been revealed to be effective in the inhibition of growth and reducing the number of foodborne pathogens, such as *Salmonella* spp., *Escherichia coli* O157:H7, *S. albus*, *Bacillus subtilis*, *Salmonella typhimurium*, *Shigella dysenteriae*, and *Listeria monocytogenes* [29,30]. In addition, the association of several compounds present in EOs provides antioxidant properties, mainly due to the presence of phenolic compounds as major components [4]. Antioxidant compounds pose the ability to delay or inhibit the oxidation of lipids and other molecules by inhibiting the initiation or propagation of oxidation chain reactions [9]. Studies present that the essential oils extracted from cinnamon, nutmeg, clove, basil, parsley, oregano, and thyme are characterized by the most important antioxidant properties [31]. This paper’s aim is to evaluate the current knowledge on the applicability of EOs in food preservation, and how this method generally affects technological properties and consumers’ perceptions. Further, the scientific literature from the last five years is investigated for evaluating the latest updates in the field.

## 2. Active Packaging Systems with Essential Oils Addition

Food packaging protects food from environmental factors such as UV light, oxygen, water vapor, pressure, and heat and improves shelf-life and food safety. Active packaging interacts with the product and, although it may have some synthetic compounds, the addition of EOs and biodegradable materials has made industrial producers determined to use it as an environmental-friendly packing method, with increased consumer acceptability [32]. Delivery systems may be particulate (lipid-based—emulsions, emulgels, nanostructured lipid carriers, and solid-lipid nanoparticles; self-assembled—liposomes, phytosomes, ethosomes, and niosomes; polymer-based—micro/nanoparticles, micro/nano-capsules, and nanofibers), macroscopic carriers (film and sponge), or molecular inclusion complexes (cyclodextrins and nontoxic cyclic oligosaccharides) [33]. The EOs have a series of structural disadvantages that limit their industrial use: volatility (influenced by heat, air, illumination, or radiation); oxidizability, which may lead to compounds of degradation that produce consumer harm; an intense aroma that can cover or modify the original food smell; reduced solubility (they are hydrophobic and incorporation in water-based matrices is challenging); possible interaction with nutritional constituents from food [34].

Nanotechnologies use liposome, nanoemulsion, or solid-lipid nanoparticle-based strategies to enhance water solubility and bioavailability, protect the EO, mask the unpleasant odor or taste, and assure a controlled release of active ingredients. Nano-encapsulation of EOs has a lot of benefits, but toxicological aspects of most of the nanocarriers are not fully explored and, before commercial application, consumer safety should be checked [35]. Incorporation in colloidal delivery systems (emulsion-based, surfactant-based, or biopolymer-based) improved the antimicrobial activity of EO in the food industry by increasing the aqueous solubility and mass transfer of EO [36]. *Citrus* spp. essential oils incorporated in edible films and coatings, microencapsulated biodegradable polymers, and nano-emulsion coatings were assessed for their potential in active packing with good effects as antimicrobials and antifungals and good environmental impact (by-products can be used, reducing food waste) [37], but with limited use because of their interaction with food matrices, concerns about the optimum dose, its impact upon consumer’s sensorial perception, and its potential as an allergen [38,39].

In active packaging design, it is very important to choose the appropriate coating or carrier material as a delivery system for EOs considering that the polymer needs to protect the EO, assure its controlled release, and be soluble in water, biocompatible, biodegradable, and low-cost [40]. In essential oils–antibacterial packaging (EOs–AP) systems, active ingredients diffuse through micropores of the film and are released at the food surface, assuring antimicrobial protection. Several factors affect the antimicrobial efficiency: solubility of the antibacterial agent into food, the migration rate between the EOs–AP system and food, and microbial growth kinetics that must be considered when designing the AP, so the sustained release assures a minimum inhibitory concentration (MIC) and the interaction between EO constituents and AP formulation, so the physical properties of the packaging material are not affected [41]. Polysaccharides are widely used because of their advantages: low cost, safety, accessibility, biodegradability, and biocompatibility.

Chitosan, alginate, cellulose, starches, gums, and pectin are some of the most used biopolymers [42]. Starch-based food packaging is used as an environmentally friendly packing method with the disadvantages of high permeability to water vapor transfer and bacterial or fungal spoilage. The latter may be solved by incorporation of EOs to prevent microbial contamination, but it reduces the mechanical properties of the film: reduces the cohesion forces of the polymer chain, creating a heterogeneous matrix and subsequently lowering the tensile strength and increasing the elongation at break value of the film, which must be considered when designing a product covered in starch film enriched with various EOs [43]. Chitosan is a polysaccharide with antimicrobial properties and its enrichment with EOs improves fruit preservation prone to fungal infections, depending on releasing the antimicrobial compounds from the polymer matrix, the type of EO, fungal species, and incubation temperature [44]. Hadidi et al., 2020, loaded chitosan nanoparticles with clove essential oil (CEO) with a 55.8–73.4% retention rate of EOs and obtained a better antioxidant capacity than the free CEO and high antimicrobial activity against *Listeria monocytogenes* and *Staphylococcus aureus*, being a potential delivery system to be used in active packaging [45]. Hydroxypropyl methylcellulose edible films containing nanoemulsions of wild and cultivated *Thymus daenensis* essential oil were used by Moghimi et al., 2017, for their antimicrobial effect as a food packaging system. This study showed that wild thyme had more antibacterial activity against Gram-positive bacteria (including *S. aureus*, *S. epidermidis*, *B. subtilis*, *E. faecalis*, and *E. faecium*) rather than the cultivated one, while the latter was more efficient against *Candida albicans* and Gram-negative bacteria (including *E. coli*, *S. typhi*, *S. dysenetriae*, *S. flexneri*, *A. baumannii*, and *K. pneumoniae*), both being potential active food packaging systems [46]. Arezoo et al., 2020, used sago starch films loaded with cinnamon EO (1–3%) and TiO2 (1–5%) as a bio-nanocomposite for maintaining the quality of fresh fruits and vegetables with good antimicrobial protection against *Salmonella typhimurium*, *Escherichia coli*, and *Staphylococcus aureus* [47]. Lemon waste powder obtained from byproducts of lemon juice was incorporated with 3–6% cellulose nanofibers and 1.5–3% savory EO into a nano-biocomposite film, being a promising biodegradable active packing system that contributes to waste reduction and can offer good antibacterial protection against *S. aureus*, *L. monocytogenes*, *E coli*, *Ps. Aeruginosa*, and *S. enterica* [48]. Inclusion complexes (ICs) with *Ocimum basilicum* and *Pimenta dioica* EOs and β-cyclodextrin (β-CD) were prepared as a strategy to reduce volatility and increase the release time of EOs by Marques et al., 2019, with good antimicrobial properties, superior for sachets with *Pimenta dioica*, and the potential to be used as a food preservative [49]. Coriander EO was incorporated in cyclodextrin and maltodextrin nanosponges (NSs) as a stable controlled release system with antimicrobial properties against *Campylobacter* spp., *Listeria monocytogenes*, and *E. coli*, with better antimicrobial activity for cyclodextrin NS than maltodextrin NS expressed in lower MICs than coriander EO [50].

## 3. Applicability of Essential Oils in Food Products

Considering that the EOs are perceived as an alternative eco-friendly food preservative, the interest in EOs application in the food industry and packaging has grown in recent years. The major sources relevant for industrial use are from *Alliaceae*, *Apiaceae*, *Asteraceae* (*Compositae*), *Lamiaceae (Labiatae)*, *Myrtaceae*, *Poaceae*, *Cupressaceae*, *Lauraceae*, *Pinaceae*, *Zingiberaceae*, and *Rutaceae* plant families and the extraction can be made from different part of plants: flowers, leaves, fruits, seeds, grasses, roots, rhizomes, wood, bark, gum, tree blossoms, bulbs, and dried flower buds [51]. EOs play significant roles in the food sector with a wide range of applications, mainly to prolong shelf-life and prevent oxidation. For this purpose, they are valorized by the addition to food products, either by direct mixing or in edible coatings and active packaging [32]. Several EOs have GRAS status (Generally Recognized as Safe), including basil, cinnamon, clove, coriander, ginger, lavender, menthol, nutmeg, oregano, rose, sage, and thyme EOs. The reported EO constituent comprises carvacrol, carvone, citral, cinnamaldehyde, eugenol, limonene, linalool, thymol, and vanillin, with no antagonistic human health effects [11]. In the study conducted by Macchia et al., 2022, the efficacy of pure EOs of eucalyptus, basil, cloves, thyme, pine tree, and tea tree was assessed in situ and compared to a conventional biocide based on quaternary ammonium salts. The tests revealed positive results concerning the synergy of the essential oils, showing a stronger biocide efficacy [52].

EOs extracted from ten *Apiaceous* fruits [*Pimpinella anisum* L. (anise), *Carum carvi* L. (caraway), *Apium graveolens* L. (celery), *Coriandrum sativum* L. (coriander), *Cuminum cyminum* L. (cumin), *Anethum graveolens* L. (dill), *Foeniculum vulgare* L. (fennel), *Petroselinum crispum* L. (parsley), *Daucus carota* L. var. *sativus* (yellow carrot), and *Daucus carota* L. var. *boissieri* (red carrot)] were studied for their antimicrobial effects by Khalil et al., 2018; among them, coriander, caraway, and cumin EOs were the most potent bactericides [53]. In recent decades, bacterial resistance has been one of the main threats to global health [54]. Plant secondary metabolites, such as EOs, phenolic compounds, alkaloids, lectins/polypeptides, and polyacetylenes, represent an alternative to reduce this issue [18]. The main pathogens involved in the spoilage of food products and causing foodborne diseases are *E. coli*, *Clostridium* spp., *Salmonella* spp., *Campylobacter* jejuni, *Aeromonas hydrophila*, *S. cerevisiae*, *Penicillium expansum*, and *Listeria monocytogenes* [55]. Fernandez-Lopez and Viuda-Martos completed a review in 2018, and they demonstrated the wide application of EOs by taking into account the number of scientific publications found when the words “antioxidant” (1920 results), “antimicrobial” (2473 results), or both (973 results) were used as keywords along with EOs [56]. EOs are used as biopreservatives in all food types with various applications in the food segment for meat, fish, seafood, bread, grains, milk, dairy products, fruits, and vegetables (especially cut products) to enhance the quality and safety of products. The recent application of essential oils in the food industry is presented in Table 1.

Some properties of EOs, such as the low-level water solubility, the high volatility, and their powerful odor, can curb their use as food preservatives in the food industry [87,88]. The antioxidant activity of biofilms depends on the antioxidant action of the EOs and the film’s oxygen permeability. Adding EOs into edible films can boost their antimicrobial capability, and the edible film’s efficacy against microbial growth will vary based on the oil’s chemical characteristics and microorganism type [89]. Incorporating EOs and plant extracts into edible natural biopolymers represents one of the newest research trends of the food packaging industry [90,91]. Their incorporation into packaging materials and coated films and straight into the food matrix as emulsions, nanoemulsions, and coatings are part of newer applications [92].

Elevated contents of EOs can result in the transmission of off-flavors to the end product. Moreover, as EOs are lipophilic, it is difficult to disperse them in water-based washing solutions. Given this, an alternative was developed, namely their incorporation in nanoemulsions [93]. Plant EOs are the organic and natural option to other preservatives with significant antimicrobial, antifungal, and antiviral properties when used to obtain innovative and healthy packaging. Consumers want to be assured that the packaging has the proper attributes for ensuring food quality and freshness and preventing microbial contamination.

Until now, there are few patents regarding the addition of EOs in active food packaging [94]. The subject of WO patent 2013032631 A1 is the encapsulation of aromatic compounds of EO within gelatin capsules with potential application in beverage packaging [95]. Another patent concerning a plastic film with EO incorporated was created to protect and preserve horticulture products and foods against insects (US 20040034149 A1) [96]. The subject of WO patent 2013084175 A1 is a material of animal origin with EO isolated from *R. officinalis*, *Citrus limon*, and *Vitis vinifera*, used to pack fresh food and to inhibit the development of biogenic amines [97]. Another patent WO 2006000032-A1 regards an antimicrobial packaging material for food products containing from 0.05% to 1.5% by weight of a natural EO [98]. Patent number WO 2007135273 A3 refers to materials based on (non)woven fibers for food preservation, coated with a matrix containing biodegradable polymers for controlled release of volatile antimicrobial agents [99].

### 3.1. Fruits and Vegetables

Fruits and vegetables are acknowledged increasingly for their health benefits, mainly due to their nutrients and fiber content, while being characterized by a short shelf-life due to weight loss and decay, caused mainly by fungal activity. *Botrytis cinerea* appears very frequently in fruits during the postharvest period. Strategies enhancing the shelf life of fruits and vegetables cover the combined application of EOs into biofilms or MAP (modified atmosphere packaging) [100]. To benefit from fresh fruits and vegetables in a time-convenient manner, the food industry has launched the concept of “fresh-cut food”, which implies that fruits or vegetables were physically altered but remained fresh. In addition to the health benefits, fresh-cut food is appealing (fresh-like appearance, taste, and flavor) and ready-to-eat [101]. Among the advantages of this segment of products, there is the fact that they require less time to be prepared and consumed. These products are very time-convenient for customers as they are already cleaned and trimmed, reducing the preparation time. While fresh-cut fruits and vegetables (FCFV) hold some benefits (convenient, portable, and easy to eat) over the whole product, the industry and the processes followed to preserve their freshness and quality (free of pathogenic microorganisms) face considerable challenges [102]. While the health benefits of fresh-cut products have persuaded consumers to increase their intake, studies have revealed that the number of illnesses caused by *Salmonella enterica*, *Escherichia coli*, *Shigella* spp., *Campylobacter* spp., *Listeria monocytogenes*, *Staphylococcus aureus*, *Yersinia* spp., and *Bacillus cereus* has also increased [103].

There is an increasing trend in the fresh fruits and vegetable packaging area to substitute the traditional petrochemical-based packaging films with environmentally friendly and biodegradable materials, such as edible coatings. Most used compounds for obtaining edible coatings comprise chitosan, starch, cellulose, alginate, polyvinyl alcohol (PVA), carrageenan, zein, gluten, whey, carnauba, beeswax, and fatty acids [104]. Another essential benefit of using edible coatings is reducing synthetic waste, given that edible coatings are made up of biodegradable material. For example, biodegradable green packaging can be obtained from tropical plants (neem, nutmeg, turmeric, cinnamon, and lemongrass) and some of their by-products (seeds, husks, kernels, and peels) with beneficial bioactive properties [105]. In fresh or fresh-cut products such as fruits and vegetables, several parameters are evaluated to assess the EO coating efficiency: chemical properties (total soluble solids, vitamin C content, and titratable acidity), weight loss and firmness, respiration properties, microbiological properties, antioxidant properties (total phenolic content and DPPH radical scavenging activity), and impact upon sensorial characteristics (flavor, taste, visual appearance, and color) [106].

There is great interest in using EOs as effective agents against antimicrobial contamination in minimally processed food [21]. The EOs possessing the most potent antibacterial properties have antimicrobial agents such as carvacrol, eugenol, and thymol [16]. The antimicrobial action of EOs in vegetables and fruit-based dishes are influenced positively by low storage temperature and pH reduction [107]. In several studies, the researchers used the EOs obtained from orange, lime, *Litsea cubeba*, clove, oregano, spearmint, cinnamon, limonene, and thyme in developing packaging for fruits and vegetables with promising antimicrobial results [108,109,110,111,112,113]. There are various studies regarding antimicrobial packaging based on a blend of chitosan, gelatin, carboxymethyl cellulose—CMC, polylactic acid (PLA), and essential oils (e.g., *Mentha spicata* oil, Red thyme, and *Zataria multiflora*) designed to expand the shelf life and stability of strawberries [61,114,115,116,117,118]. Citronella EO was loaded on a nanocomposite film made of chitosan/ZnO/Ag, and the antimicrobial effect upon S. aureus, *E. coli*, and *C. albicans* was studied on grapes. The synergistic effect of EO, ZnO, and Ag increased the shelf life of packaged white grapes to 14 days with very good visual appearance and antimicrobial control [119].

In others studies, a mixture of eugenol, thymol, and carvacrol has been tested on grapes and showed that counts for yeasts and molds were significantly reduced, as well as the lower occurrence of berry decay [120,121,122]. The study conducted by Waithaka et al., 2017, demonstrated that EOs extracted from rosemary and eucalyptus have the capability of controlling passion fruit fungal pathogens such as *Alternaria* spp., *Fusarium* spp., *Colletotrichum* spp., and *Penicillium* spp. [123]. Munhuweyi et al., 2017, investigated the inhibitory effects of chitosan EOs, with different concentrations of lemongrass, cinnamon, and oregano oils, using vapor emission and direct coating against *Botrytis* spp., *Penicillium* spp., and *Pilidiella granite* pathogens of the pomegranate fruit. The results showed that chitosan film incorporated with oregano EO had the highest antifungal activity [124]. Hoseini et al., 2019, used a chitosan-based coating incorporated with savory (*Satureja hortensias* L.) and tarragon (*Artemisia dracunculus* L.) EOs in prolonging kumquat’s shelf time by 30 days. The active packaging was efficient in reducing weight loss during storage and maintaining vitamin C in fruits with good sensory acceptability [125]. Mango fruits were covered with carboxymethyl chitosan (CMCS)-pullulan edible films enriched with 4–12% galangal EO, the best mechanical properties of the film being with a 2.5:2.5 CMCS-pullulan ratio. The most effective film was with 8% galangal EO addition, providing 15 days of shelf life for mango fruits while maintaining its properties (firmness, acidity, soluble solids, and weight) [126]. In the work of Velázquez-Contreras et al., 2021, active packages filled with thymol or carvacrol complexed in 3-cyclodextrins (ß-CDs) showed the effectiveness of these monoterpenes in improving the shelf life of berries fruits [127]. Dwivedy et al. (2018) showed the in situ potency of *Illicium verum* EO in the complete protection of pistachio seeds from fungal and aflatoxin B1 contamination [128]. *Origanum majorana* essential oil encapsulated into chitosan nanoemulsion caused in situ inhibition of lipid peroxidation and AFB1 production in maize without altering their sensory properties [129].

### 3.2. Cereal-Based Products

In agriculture, crops are affected by fungal infections, and the antifungal effects of some EOs are already valorized in industrial settings: rosemary, clove, thyme, sesame, and garlic are just some of the natural sources of EOs included in commercial products for crop protection [6]. Lee et al., 2017, used garlic, onion, ginger, fennel, and black pepper EOs on multilayered rice films made from rice flour and LDPE (low-density polyethylene) as a potential insecticide against *Plodia interpunctella* larvae infestation on stored cereals. Garlic and onion EO had the best anti-insect effect, and allyl disulfide, extracted from garlic, was further used for brown rice packaging with good sensory evaluation [130]. EOs loaded on lipid nanocarriers (nanoemulsions, solid-lipid nanoparticles, and liposomes) inhibit food pathogens by changes in the cellular morphology and functionality (ribosomes and cytoplasm), interruption of the electron transport chain, and inhibition of bacterial toxin release, and they can be used for preserving grains and flours [131]. *Oregano majorana* EO encapsulated in a chitosan-based nanoemulsion inhibited lipid peroxidation and AFB1 (aflatoxin B1) production in maize without affecting the sensorial properties, being a promising antifungal agent for crops [129]. Cinnamon and clove oils were added to a pectin-based film and applied to bread to reduce spoilage and increase shelf life. Cinnamaldehyde from cinnamon EO and eugenol from clove EO offered good antifungal protection against *Penicillium* and *Aspergillus*, increasing bread shelf life by 4 days [132]. Das et al., 2019, concluded that *Coriandrum sativum* essential oil (CSEO) and encapsulated CSEO nanoemulsion have a strong potency in the suppression of in situ Aflatoxin B1 production in stored rice seeds than a fungal infestation [71].

Worldwide, there is a global concern about synthetic insecticides’ negative impact on ozone, environmental pollution, toxicity to nontarget organisms, and pesticide residues. The negative effects of synthetic pesticides have amplified the need for effective and biodegradable pesticides. EOs and their phytoconstituents have been reported as natural agents against insect pests and could be applicable to the management of insect pests to decrease ecologically detrimental effects [6]. Another study conducted by Yang et al., 2020, evaluated 28 essential oils for their attraction–inhibitory activity against the adults of *S. zeamais*. Amongst the four most active oils (cinnamon, tea tree, ylang ylang, and marjoram oils), cinnamon oil was the most active in both contact/residual and fumigant bioassays and exhibited strong behavioral inhibitory activity [133]. The findings of Bendini et al., 2020, indicate that the use of EOs from mandarin (*Citrus reticulata*) and tea tree (Melaleuca alternifolia) for the post-harvest protection of small fruits against Fruit Fly *Drosophila suzukii* is feasible, provided that the EOs have been selected not only for their bioactivity against the insect pest but also for their affinity with the consumers’ sensorial system [134].

### 3.3. Dairy Products

In cheese, the main concern is microbial spoilage due to fungi and bacteria. EO incorporation in cheese active packaging has benefits such as increasing the shelf-life of cheese products, but a series of limitations occur: cheese proteins interact with phenolic compounds from EO; fats surround hydrophobic constituents of EO, which may interfere with their antimicrobial activity; the physical structure of cheese may limit EO availability to microbial cells; the intense aroma of EO may interact with the cheese taste [135]. Oregano or garlic EO was added to a whey protein (WPI) edible film and applied to Kasar sliced cheese to provide microbial reduction during cheese storage. Garlic EO-WPI was less effective than oregano EO-WPI on microbial protection, assuring 15 days of shelf life and contributing to reducing food waste [136]. Mahcene et al., 2021, used sodium alginate-based edible films enriched with rosemary, basil, white wormwood, and pennyroyal EOs as preservatives for homemade cheese. They assured protection against weight-loss, hardness, discoloration, flavor, and texture changes and prevented microbial spoilage, with good consumer acceptance, and with the film enriched with basil being most preferred [137]. In the study conducted by Moghazy et al., 1–2% of encapsulated thyme EO was added to a liposomal chitosan emulsion and applied on Karish cheese to increase shelf life. Thymol and p-cymene identified in thyme EO were responsible for 4 weeks of antimicrobial protection against aerobic bacteria, psychrotrophic bacteria, and yeast and mold. In addition, 2% chitosan solution stabilized the liposomal emulsion for 2 months at 4 °C and, as a result, coating with this emulsion increased the shelf life of Karish cheese from 2 to 4 weeks [138]. *Carum copticum* EO was included in a composite film with Pectin/Nanoclay (montmorillonite) and β-Carotene and was applied on butter; the antioxidant and antimicrobial properties, and color changes were assessed. The film containing 0.05% EO and 0.03% β-Carotene assured protection against *B. cereus*, more than for *E. coli*, and had good antioxidant properties with the highest oxidative stability and least color change during 60 days of storage. The film changed color from orange to yellow when butter oxidized and could be considered expired [139].

### 3.4. Eggs

Pires et al., 2020, covered eggs with a series of rice protein coatings enriched with 1% tea tree (*Melaleuca alternifolia*), copaiba (*Copaifera langsdorffii*), or thyme (*Thymus vulgaris*) EO to increase the shelf life of eggs. After six weeks of storage, covered eggs had better internal properties than uncovered eggs, in terms of weight loss, albumen pH, and yolk index, probably due to the hydrophobicity of the rice coating and the lipophilic characteristics of EO that provide a barrier to mass and oxygen loss [140]. Basil EO (BEO) and beeswax were incorporated in a chitosan-based emulsion and used for egg coating. The proportion of beeswax and BEO that assured the best coating properties by increasing the consistency coefficient of the emulsion, decreasing the droplet size, improving the stability by forming a multi-layer adsorption at the oil–water interface, and having bacteriostatic properties was, respectively, 0.5% and 1%. Shelf life was extended until day 35, with no height loss (eggs remaining in the AA category) and a better yolk index than the control group [141].

### 3.5. Meat and Meat Products

Rosemary EO was incorporated in a bio-nanocomposite based on chitosan and montmorillonite (MMT) to develop an active packaging for fresh poultry meat. It increased the storage time up to 15 days by inhibiting lipid peroxidation and discoloration and assuring microbiological safety, but MMT interacted with the plastic wrap used for primary packing. In an industrial setting, either the fresh poultry meat producer chooses another way of primary packing, or the active packaging should be consisted of chitosan films enriched with rosemary EO, without MMT [142]. Vital et al., 2018, covered beef meat with an alginate coating enriched with 0.1% rosemary EO or oregano EO and carried out a sensory analysis of steaks to determine consumers’ acceptability of EO coating. Although both EOs have a strong flavor, steak coated with oregano EO was preferred by the panelists, showing the importance of choosing the appropriate flavor when designing an active packaging for meat [143]. Thyme EO was included in a β-cyclodextrin complex and further incorporated into ε-polylysine nanoparticles (TGPNs). Gelatin nanofibers containing TGPN were engineered and loaded on fresh chicken meat, and the antimicrobial activity on the *C. jejuni* was evaluated. The TGPNs packaged chicken samples possessed good antimicrobial protection without a negative impact on color, texture, and sensory evaluation, being a promising prospect in meat preservation [144]. *Ferulago angulata* essential oil (FAEO) was used to enrich a gelatin-chitosan edible film applied to turkey meat, and 0.5% FAEO was the concentration that assured the best film stability, ensuring microbial growth inhibition and increasing the storage time to 15 days [145]. The casein-maltodextrin encapsulated with thyme (*Thymus vulgaris*) EO showed antioxidant and antimicrobial activity against *Staphylococcus aureus*, *Escherichia coli*, *Listeria monocytogenes*, and *Salmonella Typhimurium* tested in vitro and against thermotolerant coliforms and *Escherichia coli* in situ, showing potential for application as a natural preservative in hamburger-like meat products [146].

### 3.6. Seafood

One of the biggest challenges for the seafood industry is the short shelf life of fresh marine food products due to microbial contamination and biochemical spoilage caused by lipid/protein oxidation, a problem of great economic concern. Seafood products are highly perishable due to a large amount of polyunsaturated fatty acids and the strong activity of endogenous enzymes [147,148]. Fresh marine food products may be subjected to microbial contamination or biochemical spoilage during handling, processing, and storage. Therefore, the need for improving the quality and increasing the shelf life of fresh seafood products is currently increasing [149]. Therefore, the need for improvement in the quality and increased shelf life of fresh seafood products is currently increasing. The need of increasing the shelf life of fresh fish and other seafood products without compromising on the quality and safety has led to an increased number of studies focusing on the application of natural antioxidants to retard lipid and protein oxidation [150]. Modern consumers prefer the use of antioxidants from natural sources due to the lower risk of cardiovascular diseases and cancer [25]. Most of the natural antioxidants used for seafood preservation belong to the group of phenolic compounds naturally found in plants such as phenolic acids, anthocyanins, and flavonoids. As these compounds originally protect plants from oxidative stress, they can successfully be used as a source of natural antioxidants for the preservation of marine food products. The use of natural antioxidants in the food industry including seafood processing is often performed by the application of EOs [151].

EOs exhibit strong antioxidant, antibacterial, antiviral, and antifungal properties and are considered safe additives, which enable their application in the seafood industry. Those derived from spices and certain herbs are also found to possess antimicrobial activity and are, therefore, used in the seafood industry as natural agents for extending the product shelf life by retarding microbial and biochemical spoilage [27]. Recently, research focusing on the application of EOs for chilled stored fish gained rapidly growing popularity due to very promising results. The fish processing industry has shown a growing interest in using EOs as bio-preservatives and antimicrobials for chilled-stored seafood products [25]. The most commonly used fish species for EOs application studies were salmon, rainbow trout, grass carp, and common carp [65,66,67,69,86] and the most popular EOs were oregano, dill, thymus, rosemary, clove, cinnamon, and mint [25]. The EOs have been applied to fish using different methods, i.e., directly through spraying, immersion, pipetting, or evaporation, as well as bulk EO or EO emulsion, coupled with other stabilization methods such as modified atmosphere packaging (MAP), active film, additives, and various pre-treatments [25]. The EOs inhibitory effect of the microbial population on the spoilage parameters in chilled-stored seafood varies among EO varieties. However, it is possible to decrease the bacteria load with lower doses of EOs when applying hurdle technologies such as modified atmosphere packaging [67]. At the same time, large fluctuations in an EO’s inhibitory effects on microorganisms were observed even at the same dose of EOs. This phenomenon can be explained by different compositions (fat, water, and protein content) among various fish species, EO types, and methods of using EOs [25]. However, generally, EOs are very effective against microbial communities in chilled-stored fish, as well as biogenic amines (such as histamine, putrescine, and cadaverine). Moreover, a number of volatile organic molecules related to fish spoilage are suppressed by the EOs [67]. These studies confirm the effectiveness of EOs in reduction in microbial and biochemical spoilage in chilled-stored fish. *Citrus* oils have also been shown to be effective in fish products; the microflora of carp skin, gill, and intestines is reduced by citral and linalool at 20 °C for 48 h [152].

## 4. Conclusions and Future Perspective

The results reported in this review provide updated information concerning the use of EOs in the food sector, which are an alternative to synthetic antioxidants for conventionally produced foods. The available literature indicates that essential oils have been used with success against a wide range of foodborne pathogens. Some properties of EOs, such as the low-level water solubility, the high volatility, and their powerful odor, can curb their use as food preservatives in the food industry. Several methods have been considered to overcome this problem. Therefore, a possible solution would be their inclusion in an edible coating. Seeking to lengthen the shelf-life and to increase a product’s value, but also to strengthen consumer confidence in processed food products, EOs can add various properties to films and coatings, such as antioxidant and/or antimicrobial characteristics, varying EO’s compounds and their interactions with the polymer matrix. However, the use of EOs has several limitations: most of the studies were performed in vitro and in vivo; there may be different outcomes, due to their high volatility, and low water solubility; volatile oils also impart a strong flavor and consumers may reject them; the yield is influenced by the variety of plants, extracting techniques, harvesting season, or geographical regions. This could generate low reproducibility of a process in an industrial setting, and the cost, selectivity, and safety could be a limitation in their use.

Mainly, EOs have been successfully used for fruits and vegetables, meat, fish, and dairy products preservation. However, regarding fruits and vegetables, the disadvantages mentioned above make their applications limited. This issue is primarily given by the significant amount of EOs’ volatile compounds, which mask the natural flavor of fruits and vegetables. A potential solution might be to study further the use of composite films based on a similar product matrix as seen in the case of adding lemon EOs to biofilms used for packing citric fruits. These methods would lessen the olfactory impact of EOs on fruits and vegetables; in that sense, encapsulation and nanotechnology are promising tools.

A highly researched topic is developing new natural edible films and coatings having an inherent antimicrobial activity or one obtained through the addition of EOs. The literature consulted shows an increased interest in this topic. There are several reviews in the field, but there are few research/original articles, and they have been published recently (since 2015). Research results showed that the new active packaging was more successful in vitro than in vivo (i.e., with real-food products), pointing to a need for continued research. There are regulatory constraints regarding the accepted daily intake of EOs and some of their components; therefore, for using them appropriately in food products, a daily intake survey should be prepared for the health authority’s evaluation. Moreover, a deeper knowledge of the effects and the role of EOs on human health needs to be better investigated. Future studies should be carried out on their mode of action and their possible toxicological effects in order to optimize their potential uses and increase bio-accessibility.

The utilization of essential oil in active food packaging seems to be a realistic solution to prolong the shelf life of food products and maintain their safety, quality, and integrity. Although several patents have been achieved due to the positive result of EOs incorporation into food packaging, there is no information available in the literature about their commercial used.

## Figures and Tables

**Figure 1 plants-11-02488-f001:**
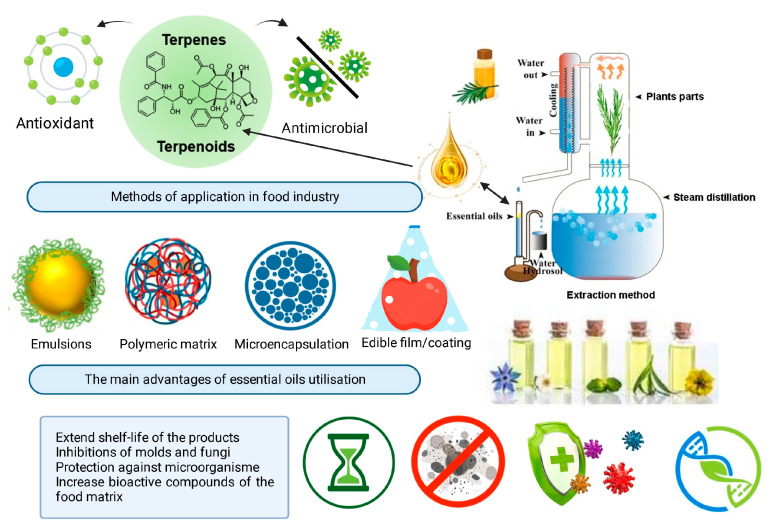
The steam distillation technique and principal methods of essential oils application in the food industry.

**Table 1 plants-11-02488-t001:** The application of essential oils in the food industry.

Plant Essential Oil	Major Active Ingredients	Concentration of EO in Product	Type of Product	Target Microorganism	ModernTechnology (Film/Coating)	Sensory Acceptability Tested Attributes	Effect of EO Addition	Examples of FoodProducts	References
*Rosmarinus officinalis* (Rosemary)	1,8-Cineole (27.5%), α-Pinene (21.2%), Camphor (12.8%), Camphene (10.3%), β-Pinene (8.9%), and Borneol (3.0%)	1–2% 0.5–1% 0.2–1%	Meat and meat products	*E. coli*, *S. enteritidis*, *L. monocytogenes* and *S. aureus* *Pseudomonas* spp., *Enterobacteriaceae*, lactic acid bacteria (LAB), yeasts	Coating with chitosan/chitosan-caseinate Cellulose nanofiber/whey protein matrix containing TiO_2_ particles (1%) Whey protein film Chitosan coating	Color measurement Odor, taste, and overall perception	Extended shelf life from 6 to 12–15 days Extended shelf life with good sensory acceptability to 8 days Inhibition of mold formation for 9 days with good sensory acceptability	Fresh minced	[57]
								Chicken breast	[58]
								Lamb meat	[59]
								Sukuk	[60]
*Mentha spicata* (Mint)	Carvone (78.76%), Limonene (11.50%), and β-Bourbonene (11.23%)	0.1–0.2%	Fruits	*L. monocytogenes*	Carboxymethyl cellulose (CMC) and Chitosan (CH) coating	Appearance, color, texture, and overall acceptability	Extended shelf life at least 12 days	Fresh strawberries	[61]
*Origanum vulgare* (Oregano)	Carvacrol (58.30%), Linalool (9.09%), and γ-Terpinene (6.01%)	2% 0.5–1% 1.5–2.5% 0.1% 112 ± 13 mg/fillet 2%	Meat and meat products Dairy products Fish Fish Fish	*E. coli*, *S. aureus*, and *S. typhimurium* *Total viable counts (TVC)* *Aeromonas* spp. *Shewanella* spp. Biogenic amines: putrescine, cadaverine, spermidine, spermine, histamine, tyramine, tryptamine and phenylethylamine Microbial growth (total culture plate count). VOCs as indicators of microbial growth.	Nanoemulsion-loaded pectin (PEC) edible coating Chitosan film Nanoemulsion-based edible coating with mandarin fiber Immersion in 0.1% (*v*/*v*) oregano EO emulsion for 30 min at room temperature and then air packaging in polyvinyl chloride bags followed by chilled storage at 4 °C UV-C-treated at 121 and 243 mJ/cm^2^, vacuum-packed, and chill-stored at 3.5 °C MAP + oregano EO solution by spraying followed by chilled storage at 4 ± 1 °C for 16 days	Texture and color measurement Color, odor, and overall acceptability Color, odor, texture, and general acceptability with 9-point hedonic scale test No sensory analysis performed Color instrumentally. Volatile compounds by GC–MS, SIFT–MS, and SESI–MS instead of sensory analysis	Extended shelf life up to 20–25 days Extended the shelf life up to 24 days *Aeromonas* spp. reduced from 48.76% to 19.1%, *Shewanella* spp. reduced from 9.64% to 1.7% Decrease in Hx up to 1.2 μmol/g, TVBN decreased up to 10.5 mg/100 g, prolonging the shelf life to 8 days. A microbiological shelf-life extension of 2 days Biogenic amines decreased: putrescine up to 20 mg/kg and cadaverine up to 38 mg/kg. Prolonging the shelf-life by 6 times Inhibited bacterial growth up to day 8 of storage	Fresh pork loin	[62]
								Beef meat	[63]
								Low-fat cheese	[64]
								Grass carp (*Ctenopharyngodon idellus*)	[65]
								Common carp (*Cyprinus carpio*)	[66]
								Atlantic salmon (*Salmo salar*)	[67]
*Syzygium aromaticum* (Clove)	Eugenol (56.06%), Caryophyllene (39.63%), and α-Caryophyllene (4.31%)	1% 0.5 %	Meat and meat products Fish	*S.typhimurium*, *E. coli*, *L. monocytogenes*, *S. aureus* *Shewanellaceae* spp., *Pseudomonadaceae* spp., and *Flavobacteriaceae* spp.	Sodium alginate and emulsifiers Chitosan-based coating	n/a Off-odor substances with bitter taste measured as inosine	Better antimicrobial protection for *S. aureus* than nitrites Reduced total viable counts and slowed the accumulation of harmful substances including total volatile basic nitrogen, trimethylamine nitrogen, and hypoxanthine and and decreased K-value up to 11 days	Beef burger	[68]
								Grass carp (*Ctenopharyngodon idellus*)	[69]
*Anethum Graveolens* *(Dill)*	α-phellandrene dill ether limonene p-cymene	2%	Fish	Microbial growth (total culture plate count). VOCs as indicators of microbial growth	MAP + dill EO solution by spraying followed by chilled storage at 4 ± 1 °C for 16 days	Color instrumentally. Volatile compounds by GC–MS, SIFT–MS, and SESI–MS instead of sensory analysis	Inhibited bacterial growth up to day 8 of storage.	Atlantic salmon (*Salmo salar*)	[67]
*Thymus vulgaris* (Thyme)	p-Cymene (11.67–15.51%), Limonene (7.14%), Thymol (53.57–56.16%), and Carvacrol (5.47–6.93%)	0.125–0.625 g 0.5–1% 0.2–1%	Bakery products Meat and meat products	*E. faecium*, *E. hirae*, *E. coli*, *S. typhimurium*, *S.aureus*, *P. aeruginosa*, *C. albicans*, *A. niger*	Microencapsulation by complex coacervation Chitosan film	Color, odor, taste, and overall acceptability	Doubled storage time (from 15 to 30 days) without molding Increased meat shelf life to 25 days Inhibition of mold formation for 9 days with good sensory acceptability	Cake	[70]
								Beef meat	[63]
								Sucuk	[60]
*Thymus mongolicus Ronn.* (Thyme)	Thymol (72.51%) and Carvacrol (11.04%) Caryophyllene (8.99%)	0.1% 112 ± 13 mg/fillet	Fish Fish	*Aeromonas* spp. *Pseudomonas* spp. *Shewanella* spp. Biogenic amines: putrescine, cadaverine, spermidine, spermine, histamine, tyramine, tryptamine and phenylethylamine	Immersion in 0.1% (*v*/*v*) thyme EO emulsion for 30 min at room temperature and then air packaging in polyvinyl chloride bags followed by chilled storage at 4 °C UV-treated, vacuum-packed, and chill-stored at 3.5 °C	Color, odor, texture, and general acceptability with 9-point hedonic scale test No sensory analysis performed	*Aeromonas* spp. reduced from 48.76% to 38.2%, *Shewanella* spp. reduced from 9.64% to 9.17%. Decrease in Hx up to 0.7 μmol/g, TVBN decreased up to 12.5 mg/100 g, prolonging the shelf life to 8 days. Biogenic amines decreased: putrescine up to 19 mg/kg and cadaverine up to 37 mg/kg Prolonging the shelf-life by 5 times	Grass carp (*Ctenopharyngodon idellus*)	[65]
								Common carp (*Cyprinus carpio*)	[66]
*Coriandrum sativum* (Coriander)	Linalool (65.18%), Geranyl acetate (12.06%), and α-Pinene (4.76%)	0.06–0.3 g	Cereals and seeds	*Aspergillus* spp., *Alternaria alternata*, *Penicillium* spp., *Mycelia sterilia*, *Fusarium poae* and *F. oxysporum*	Chitosan-based nanomatrix	n/a	85% antifungal protection	Rice	[71]
*Zingiber Officinale Roscoe* (Ginger)	Zingiberene + Zingiberol (38.9%), Ar-curcumene (17.7%), and β-Sesquiphellandrene+β-Bisabolene (11%)	0.5–1%	Meat and meat products	*Pseudomonas* spp., *Enterobacteriaceae*, lactic acid bacteria (LAB), yeasts	Whey protein film	Odor, taste, and overall perception	Extended shelf life with good sensory acceptability to 8 days	Minced lamb meat	[59,72]
*Ocimum basilicum* L. (Basil)	Linalool (41.3%), 1,8-Cineole (9.6%), (Z)-Isoeugenol (5.9%), 1-Epi-cubenol (4.8%), α-Transbergamotene (4.6%), and (Z)-Anethol (3.2%)	500–1000 µL L^−1^ 1%	Fruits Meat and meat products	*B. cinerea* *B. thermosphacta*, *E. faecalis*, *C. maltaromaticum*, *E. coli* and *S. salivarius*	Aloe vera gel coating Chitosan film with EO microcapsules	External visual aspect	Extended 4 °C storage to 12 days with good external visual aspect Increased storage time to 10 days	Fresh strawberries	[73]
								Sliced cooked ham	[74]
*Zataria multiflora Boiss* (Shirazi thyme)	Thymol and Carvacrol	0.3–0.5% 0.5–1%	Nuts Meat and meat products	aerobic mesophilic bacteria mold and yeast	Sodium alginate coating Corn starch films fortified with cinnamaldehyde	Taste, odor, color, and overall acceptability	Extended shelf life to 39 days Extended storage to 20 days	Pistachios	[75]
								Fresh ground beef patties	[76]
*Nigella sativa* (Black cumin)	Carvone, D-Limonene, α-Pinene, and p-Cymene	1%	Meat and meat products	*S. aureus* *E. coli*	Chitosan/alginate multilayer films	Color determination	Extended 4 °C storage to 5 days with good external visual aspect	Chicken breast meat	[77]
*Illicium verum* *(Star anise)*	*trans*-Anethole (90.28%), d-Limonene (3.75%), and *4-Allylanisole* (1.51%)	0.1%	Fish	*Aeromonas* spp. *Pseudomonas* spp. *Shewanella* spp.	Immersion in 0.1% (*v*/*v*) star anise EO emulsion for 30 min at room temperature and then air packaging in polyvinyl chloride bags followed by chilled storage at 4 °C	Color, odor, texture, and general acceptability with 9-point hedonic scale test	*Aeromonas* spp. reduced from 48.76% to 36.5%, *Shewanella* spp. reduced from 9.64% to 7.28%. Decrease in Hx up to 0.5 μmol/g, TVBN decreased up to 13.5 mg/100 g, prolonging the shelf life to 8 days	Grass carp (*Ctenopharyngodon idellus*)	[65]
*Pimpinella anisum* L. (Anise)	Anethole (74.40%), Thymol (11.44%), γ-Terpinene (4.61%), D-Limonene (2.06%), and Estragole (1.87%)	0.5–2%	Meat and meat products	*Ps. Aeruginosa*, *S. aureus* *E. coli.*	Chitosan film	Odor and taste with five-point hedonic method (on cooked burgers)	Extended storage to 12 days	Chicken burger	[78]
*Pimpinella saxifraga* (Burnet-saxifrage)	Anethole (59.47%), Pseudoisoeugenol (20.15%), and p-Anisaldehyde (7.53%)	1–3%	Dairy products	*E. coli*, *Ps. Aeruginosa*, *S. typhimurium*, *L. monocytogenes*, *Micrococcus luteus*, *B.cereus*	Sodium alginate coating	Color, flavor, odor, aspect, and texture with five-point hedonic method	Extended shelf life with good sensory acceptability to 10 days	“Béja Sicilian cheese” (fresh cheese)	[79]
*Cuminum cyminum* (Cumin)	Cuminal (28.28%), p-Cymene (26.9%), γ-Terpinene (15.29%), and Phenyl glycol (14.32%)	0.5–2%	Meat and meat products	*E. coli*, *S. aureus*, *Fungi*	Shahri Balangu seed mucilage edible coating	Odor, color, and overall acceptance with 9-point hedonic scale test	Improved the shelf-life of beef to 9 days	Beef meat	[80,81]
*Paulownia Tomentosa* (Princess tree)	Geranyl, Geraniol, Nonanal, and Heptadecane	160 mg	Meat and meat products	*Pseudomonas* spp., lactic acid bacteria (LAB)	Chitosan coatings with nanoencapsulated EO	Color, odor, and overall acceptability	Prolonged the shelf life to 16 days	Ready-to-cook pork chops	[82,83]
*Satureja khuzestanica* (Marzeh-e-Khuzestani in Persian)	Carvacrol (80.55%), p-Cymene (6.43%), β-Bisabolene (3.25%), Citronellal (1.8%), and Linalool (1.35%)	1%	Meat and meat products	*Pseudomonas* spp., lactic acid bacteria (LAB)	Chitosan coatings with nanoencapsulated EO	Color and odor, with a 5-point descriptive scale	Extended antimicrobial activity during 20 days of storage	Lamb meat	[84,85]
*Citrus* *(lemon*, *orange*, *grapefruit,* and *tangerine peel)*	Limonene, γ-terpinene, and β-pinene	4%	Fish	Biogenic amines: putrescine, cadaverine, spermidine, spermine, serotonin, tyramine, dopamine, and agmatine	Application of lemon peel EO nanoemulsion by immersion, air packaging, chilled storage at 4 °C	Quality index used instead of sensory analysis: (QI) = (histamine + putrescine + cadaverine)/(1 + spermidine + spermine)	Biogenic amines decreased: putrescine up to 31 mg/kg and cadaverine up to 12 mg/kg, and histamine up to 2.4 mg/kg	Rainbow trout (*Oncorhynchus mykiss*)	[86]

## Data Availability

Not applicable.

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
