# Peer review of "An Update on Effectiveness and Practicability of Plant Essential Oils in the Food Industry"

_plants, 2022, doi:10.3390/plants11192488_

Round 1

Reviewer 1 Report

The review article is well written and the solutions presented in it - active packaging systems with essential oils addition - are very interesting. I would like to ask if any of these types of packaging are commercially used and how many of them have been patented? Please include this type of information in the text (e.g. in Conclusions).

Author Response

We thank the Reviewer for careful revision and for helpful suggestions. All observations have been considered, and the text has been modified accordingly. All modifications are highlighted in yellow in the revised version of the manuscript.

Point 1. The review article is well written and the solutions presented in it - active packaging systems with essential oils addition - are very interesting.

Response 1. Thank you for your feedback!

Point 2. I would like to ask if any of these types of packaging are commercially used and how many of them have been patented? Please include this type of information in the text (e.g. in Conclusions).

Response 2. Thank you very much for this helpful suggestion, we agreed to include the following paragraphs:

In the Applicability of essential oils in food products section:

Until now, there are few patents regarding the addition of EOs in active food packaging [86]. The subject of WO patent 2013032631 A1 is the encapsulation of aromatic compounds of EO within gelatin capsules with potential application in beverage packaging [87]. Another patent concerning a plastic film with EO incorporated was created to protect and preserve horticulture products and foods against insects (US 20040034149 A1) [88]. The subject of WO patent 2013084175 A1 is a material of animal origin with EO isolated from R. officinalis, Citrus limon, aand Vitis vinifera, used to pack fresh food and to inhibit the development of biogenic amines [89]. Another patent WO 2006000032-A1 regards an antimicrobial packaging material for food products containing from 0.05% to1.5% by weight of a natural EO [90]. Patent number WO 2007135273 A3 refers to materials based on (non)woven fibrous to food preservation, coated with a matrix containing biodegradable polymers for controlled release of  volatile antimicrobial agents [91].”

In the conclusion and Future Perspective section:

"The utilization of essential oil in active food packaging seems to be a realistic solution to prolong the shelf life of food products and maintain their safety, quality, and integrity. Although several patents have been achieved due to the positive result of EOs incorporation into food packaging, there is no information available in the literature about their commercially used."

Reviewer 2 Report

The article by L.C. Salanta and J. Cropotova is reviewing the use of essential oils in food industry mostly as alternatives to common preservatives. This update concerns various food categories as fruits and vegetables, cereals, dairy products, eggs, meat, seafood. Authors are also discussing the applicability of EOs in food products and active packaging.

Given that there is a need in food industry to replace the traditional chemical substances in food preservation and to follow the consumers as they move to healthier dietary habits, such updates are of interest and also informative due to the increasing number of related articles published and thus hard to follow.     

Overall, the article is well written, divided in easy-to-read sections and supported from adequate citations. I have no concerns over the title, the content the presentation or the references presented and I believe that it should be published.

Minor errors should be corrected during the proofing process (multiple instances of “Eos” instead of “EOs” and italicizing scientific names of bacteria or plants).

Author Response

We thank the Reviewer for careful revision and for helpful suggestions. All observations have been considered, and the text has been modified accordingly. All modifications are highlighted in yellow in the revised version of the manuscript.

Point 1. The article by L.C. Salanta and J. Cropotova is reviewing the use of essential oils in food industry mostly as alternatives to common preservatives. This update concerns various food categories as fruits and vegetables, cereals, dairy products, eggs, meat, seafood. Authors are also discussing the applicability of EOs in food products and active packaging. Given that there is a need in food industry to replace the traditional chemical substances in food preservation and to follow the consumers as they move to healthier dietary habits, such updates are of interest and also informative due to the increasing number of related articles published and thus hard to follow. Overall, the article is well written, divided in easy-to-read sections and supported from adequate citations. I have no concerns over the title, the content the presentation or the references presented and I believe that it should be published.

Response 1. Thank you for your feedback!

Point 2. Minor errors should be corrected during the proofing process (multiple instances of “Eos” instead of “EOs” and italicizing scientific names of bacteria or plants).

Response 2. Thank you very much for your observation, we have made changes throughout the manuscript.

Reviewer 3 Report

This paper's aim is to evaluate the current 85 knowledge on the applicability of EOs in food preservation, and how this method generally affects technological properties and consumers' perceptions. Further, the scientific literature from the last five years was investigated to evaluate the last updates in the field.

In abstract in vivo, in vitro etc. and the name of microorganisms must be in italic. Please check all text and correct latin name with italic.

Introduction is very well described but the last part of the mechanism needs more detailed information. In these last parts are mentioned microorganisms, only bacteria and few sentences about compounds. I think that this part needs redescription. For example more detailed mechanisms, next part about microorganisms and next part about some compounds.

Please check Table 1 and add references to each compound or product, not all references in one.

Check in all text  EOs, sometimes as Eos.

Figure 1 is about distillation and text description about edible films. This part must be chronologically corrected.

In the description, products are missing some in situ application to  products.

Also in study for food protection is some parasites, insects, a topic missing as some critical point for food quality.

Author Response

We thank the Reviewer for careful revision and for helpful suggestions. All observations have been considered, and the text has been modified accordingly. All modifications are highlighted in yellow in the revised version of the manuscript.

Point 1. This paper's aim is to evaluate the current 85 knowledge on the applicability of EOs in food preservation, and how this method generally affects technological properties and consumers' perceptions. Further, the scientific literature from the last five years was investigated to evaluate the last updates in the field. In abstract in vivo, in vitro etc. and the name of microorganisms must be in italic. Please check all text and correct latin name with italic.

Response 1. Thank you very much for your observation, we have made changes throughout the manuscript.

Point 2. Introduction is very well described but the last part of the mechanism needs more detailed information. In these last parts are mentioned microorganisms, only bacteria and few sentences about compounds. I think that this part needs redescription. For example more detailed mechanisms, next part about microorganisms and next part about some compounds.

Response 2. Thank you very much for this helpful suggestion. We avoided initially extending the introduction section in order to make it more comprehensive. In this regard, antimicrobial activity against microorganisms and antioxidant capacity of the EOs compounds have been explicitly detailed in the following sections of the manuscript. However, we agreed to include the following paragraphs in the introduction part:

“The antibacterial properties of EOs might be attributed to more than one mechanism [24]. Due to the lipophilicity of terpenes and phenolics, EOs are capable to penetrate easily inside the cytoplasm of microorganisms and deteriorate the phospholipid bilayer of mitochondria and inner membrane, resulting in increased cellular permeability followed by leakage of cytoplasmic constituents (e.g., DNA and RNA) and certain ions (Na+, K+, and Mg2+) [25]. Another mechanism is related to the distortion of lipid-protein interaction in a bacterial cell caused by lipophilic hydrocarbons naturally found in EOs affecting ATPases activity necessary for the production of ATP [26]. In addition, certain phenolics found in EOs may disrupt the electron flow, proton motive force, and cytoplasmic coagulation. All these mechanisms result in inhibition of the bacterial activity on the food surface, but also prevent active compounds from EOs to reach the inner membrane. Therefore, Gram-negative bacteria have been shown to be more resistant to EOs than Gram-positive bacteria [27]. Nevertheless, due to hydrophobic compounds present in EOs which are capable to pass through the barrier of the outer membrane, it is still possible to inactivate Gram-negative bacteria in food products. Thus, gram-negative bacteria have lower susceptibility to EOs than gram-positive ones, mainly due to their membrane characteristics that act as barriers against macromolecules and hydrophobic compounds [28]. The main bioactive compounds with significant antimicrobial activities are terpineol,thujanol, myrcenol, neral, thujone, camphor, carvone [6].

In addition, essential oils have also been revealed to be effective in the inhibition of growth and reducing the number of foodborne pathogens, such as Salmonella spp., Escherichia coli O157:H7, S. albus, Bacillus subtilis, Salmonella typhimurium, Shigella dysenteriae, and Listeria monocytogenes [29,30]. Also, the association of several compounds present in EOs provides antioxidant properties, mainly due to the presence of phenolic compounds as major components [4]. Antioxidant compounds pose the ability to delay or inhibit the oxidation of lipids and other molecules by inhibiting the initiation or propagation of oxidation chain reactions [9]. Studies present the essential oils extracted from cinnamon, nutmeg, clove, basil, parsley, oregano, and thyme are characterized by the most important antioxidant properties [31].”

Point 3. Please check Table 1 and add references to each compound or product, not all references in one.

Response 3. We agree with your suggestion and we have made changes throughout the Table 1.

Point 4. Check in all text  EOs, sometimes as Eos.

Response 4. Thank you very much for your observation, we have made changes throughout the manuscript.

Point 5. Figure 1 is about distillation and text description about edible films. This part must be chronologically corrected.

Response 5. We think your comment is accurate, the Figure 1 has been moved.

Point 6. In the description, products are missing some in situ application to products.

Response 6. We accepted your recommendation and we added further lines to describe in situ application to products. Supplementary references were added to the manuscript (ref. 52, 71, 128, 129,146).

Point 7. Also in study for food protection is some parasites, insects, a topic missing as some critical point for food quality.

Response 7. The implication of essential oils as natural agents against insect pests is a very important topic, and it deserves detail in future studies. We agreed to include the following paragraphs in order to increase the value of the paper:

Worldwide, there is a global concern about synthetic insecticides' negative impact on ozone, environmental pollution, toxicity to non-target organisms, and pesticide residues. The negative effects of synthetic pesticides have amplified the need for effective and biodegradable pesticides. EOs and their phytoconstituents have been reported as natural agents against insect pests and could be applicable to the management of insect pests to decrease ecologically detrimental effects [6]. Another study conducted by Yang et al., 2020, evaluated 28 essential oils, for their attraction–inhibitory activity against the adults of S. zeamais. Amongst the four most active oils (cinnamon, tea tree, ylang ylang, and marjoram oils), cinnamon oil was the most active in both contact/residual and fumigant bioassays and exhibited strong behavioral inhibitory activity [133]. The findings of Bendini et al., 2020, indicate that the use of EOs from mandarin (Citrus reticulata) and tea tree (Melaleuca alternifolia) for the post-harvest protection of small fruits against Fruit Fly Drosophila suzukii is feasible, provided that the EOs have been selected not only for their bioactivity against the insect pest but also for their affinity with the consumers’ sensorial system [134].”

Reviewer 4 Report

The review discusses an important topic. It comprises many references. On the topic itself there are may other reviews more specific, among others authors could view those reported for citrus essential oils. Authors should discuss them.

Potential antimicrobial uses of essential oils in food: is citrus the answer? (2008). K. Fisher; C. Phillips. Trends in Food Science & Technology. 19, Pages 156-164

https://doi.org/10.1016/j.tifs.2007.11.006

Citrus essential oils: Extraction, authentication and application in food preservation. (2019) N. Mahato, K. Sharma, R. Koteswararao, M. Sinha, E Baral, M. H. Cho. Critical Reviews in Food Science and Nutrition 59. 

https://doi.org/10.1080/10408398.2017.1384716

Antifungal Activity of Citrus Essential Oils. (2014). L. Jing, Z. Lei, L. Liet al J. Agric. Food Chem.  62, 14, 3011–3033

https://doi.org/10.1021/jf500614

 I suggest to revise the manuscript also for its form (some mistakes occur along it and the standard form is not always respected).

Author Response

We thank the Reviewer for careful revision and for helpful suggestions. All observations have been considered, and the text has been modified accordingly. All modifications are highlighted in yellow in the revised version of the manuscript.

Point 1. The review discusses an important topic. It comprises many references. On the topic itself there are may other reviews more specific, among others authors could view those reported for citrus essential oils. Authors should discuss them.

Response 1. We agree with the reviewer’s comments. Based on your suggestion we added further information to highlight the antimicrobial potential of citrus essential oils on foodstuff. The suggested references were added to the manuscript (ref. 37, 39, 152).

Point 2. I suggest to revise the manuscript also for its form (some mistakes occur along it and the standard form is not always respected).

Response 2. Thank you very much for your observation! We have made changes throughout the manuscript regarding its form.

Round 2

Reviewer 3 Report

The authors corrected the manuscript by suggestion.

Still there are minor mistakes. Please check all the text.

For example Line 16 Salmonella spp., spp. is without italics.

Line 136 Citrus with italics.

In table spp. without italics etc. 

Author Response

We thank the Reviewer for careful revision and for helpful suggestions. All observations have been considered, and the text has been modified accordingly. All modifications are highlighted in yellow in the revised version of the manuscript.

Point 1. The authors corrected the manuscript by suggestion.

Response 1. Thank you very much for your help to improve the manuscript!

Point 2. Still there are minor mistakes. Please check all the text. For example Line 16 Salmonella spp., spp. is without italics. Line 136 Citrus with italics. In table spp. without italics etc. 

Response 2. Thank you very much for your observation, we have made changes throughout the manuscript regarding textual changes.

Reviewer 4 Report

The added parts in the discussion have improved the work.

Author Response

We thank the Reviewer for careful revision and for helpful suggestions. All observations have been considered, and the text has been modified accordingly. All modifications are highlighted in yellow in the revised version of the manuscript.

Point 1. The added parts in the discussion have improved the work.

Response 1. Thank you very much for your help to improve the manuscript!